# Study on the Difference of Superhydrophobic Characteristics of Different Wood Furniture Substrates

**DOI:** 10.3390/polym15071644

**Published:** 2023-03-25

**Authors:** Xingzhou Yao, Zhangqian Kong, Feng Yang, Xinyu Wu, Yan Wu

**Affiliations:** 1College of Furnishings and Industrial Design, Nanjing Forestry University, Nanjing 210037, China; yaoxingzhou@njfu.edu.cn (X.Y.);; 2Co-Innovation Center of Efficient Processing and Utilization of Forest Resources, Nanjing Forestry University, Nanjing 210037, China; 3Beijing Institute of Fashion Technology, Fashion Accessory Art and Engineering College, Beijing 100029, China

**Keywords:** superhydrophobicity, wood, sol–gel method, coating, furniture field

## Abstract

To enhance the stability of wood and decrease restrictions on its use in the furniture industry, hydrophobic modification can be employed to confer waterproof, anti-fouling, and self-cleaning properties. The present study outlines the preparation of silica sol using the sol–gel method, followed by impregnation and chemical vapor deposition methods to modify the sol. After grafting 1H,1H,2H,2H-perfluoro-decyl trichlorosilane (FDTS), hydrophobic and superhydrophobic properties were imparted to the wood substrate. To explore the correlation between the surface properties of the wood substrate and superhydrophobic coatings, the densities, porosities, and surface roughness of various tree species were compared. The results showed that the sol–gel method successfully constructed hydrophobic coatings on different wood substrates, with six samples (poplar, elm, toon wood, paulownia, ashtree, and black walnut) achieving superhydrophobic surfaces, with densities ranging from 0.386 to 0.794 g/cm^3^, porosity ranging from 13.66 to 42.36%, roughness ranging from 4.660 to 11.244 um, and maximum water contact angle of 165.2°. Whereas beech and rosewood only reach the hydrophobic surface. Although the coatings demonstrated good resistance to water, pollutants, self-cleaning, and chemical agents, further improvements are necessary to enhance mechanical wear resistance.

## 1. Introduction

Wood stands out as the sole renewable resource within the quartet of internationally recognized raw materials, which comprises steel, cement, wood, and plastic. Due to its distinct material attributes and commendable environmental qualities, wood has gained considerable favor among the populace and finds widespread application in various domains, including construction, furniture production, and interior decoration [1]. In the present era, wooden furniture, such as office desks and chairs, wardrobes, beds, home tables, and chairs, constitutes an indispensable aspect of people’s daily lives and work routines. While providing convenience to individuals, wood furniture may also exhibit imperfections, such as cracks, discoloration, a challenging-to-clean surface, long-term water decay, and deformation, which can be attributed to the effects of temperature, humidity, and human-related factors [2]. As such, the utilization of superhydrophobic and self-cleaning surfaces in the furniture industry holds great significance, as it serves to curtail surface contamination of furniture and promote the overall health of living spaces.

Owing to their exceptional wettability characteristics [3], superhydrophobic materials find extensive application across numerous domains, such as self-cleaning [4,5], anti-freezing [6], anti-corrosion and antibacterial [7], drag reduction [8], oil–water separation [9,10], among others, and hold tremendous potential for further development. At present, the methods of preparing superhydrophobic wood include the vapor deposition method [11], the layer-by-layer self-assembly method [12], the hydrothermal method [13], and the sol–gel method [14,15]. The preparation of superhydrophobic surface materials must conform to two points: one is to construct a micro–nano rough structure on the material surface. The second is to use low surface energy substances to modify the surface with a rough structure [16,17]. Wang [18] developed a simple, efficient, and environmentally friendly approach to generating superhydrophobic wood surfaces that exhibit exceptional mechanical robustness. The entire process employs solely non-toxic silver nitrate (AgNO_3_) solution, and even when subjected to conditions such as adhesive tape peeling, abrasive paper grinding, and severe knife scraping, the wood surface consistently retains its micro/nanoparticles and endures the harsh mechanical forces. This mechanically durable superhydrophobic wood presents a promising opportunity for deployment within the civil engineering and construction domains. Jia [19] employed a straightforward alkali-driven approach that employed SiO2 nanoparticles (NP) and triethoxysilane (VTES) to fabricate highly wear-resistant superhydrophobic wood. The resulting wood features a water contact angle (WCA) of 156.6°, a rolling angle (SA) of 1.8°, and exhibits considerable wear resistance to severe mechanical damage, such as abrasive paper wear. Cao [20] achieved the successful production of multifunctional superhydrophobic wood featuring a water contact angle (WCA) of 156° and a rolling angle (SA) of 6° through the incorporation of SiO_2_ sol and superhydrophobic powder (PMHOS). The resulting superhydrophobic wood displayed excellent liquid repellency towards common fluids such as milk, soy sauce, juice, and coffee. Moreover, the superhydrophobic layer on the wood surface exhibited commendable durability even after being subjected to a series of mechanical stresses. Wu et al. [21] developed a novel method to grow silver particles on the wood surface in situ using the silver mirror method, which were then modified with stearic acid to acquire a surface with both superhydrophobic and antibacterial properties. The surface exhibited excellent mechanical properties, resistance to acid and alkali, and UV stability. The durability tests showed that the coating had good water and fouling resistance and was able to maintain its hydrophobic properties under different heat treatment temperatures. This promising approach has great potential for improving the functional properties of wood surfaces in various applications.

The current literature primarily focuses on the preparation of superhydrophobic coatings on individual wood surfaces. However, the variations in constructing the same superhydrophobic coating on different tree species’ wood surfaces require further investigation. Establishing the relationship between the fundamental properties of wood, including density, porosity, and surface roughness, and the construction of micro/nanostructured surfaces can facilitate the universal application of superhydrophobic coatings on different types of wood substrates.

In this study, we employed a variety of wood substrates, including poplar wood (*Populus tomentosa*), elm wood (*Ulmus rubra*), Chinese toon wood (*Toona sinensis (A. Juss.) Roem*.), paulownia wood (*Paulownia fortunei*), beech wood (*Zelkova serrata (Thunb.) Makino*), ashtree (*Fraxinus mandschurica Rupr.*), black walnut wood (*Juglans nigra* L.), and rosewood (*Ormosia henryi Prain*), as the base material. The sol–gel method and low surface energy modification using FDTS were employed to prepare the superhydrophobic wood surfaces. Firstly, the impact of wood properties on the fabrication of a superhydrophobic surface was evaluated by contrasting the variations in density, porosity, and surface roughness among different tree species. Subsequently, the surface functional groups, wettability, and surface morphology were assessed and evaluated using infrared spectroscopy (FTIR), scanning electron microscopy (SEM), and water contact angle test (WCA). Finally, the modified wood was subjected to an assessment of its water resistance, chemical resistance, friction resistance, pollution resistance, and self-cleaning properties.

## 2. Experimental Section

### 2.1. Materials

The poplar wood (Populus tomentosa), elm wood (*Ulmus rubra.*), Chinese toon wood (*Toona sinensis (A. Juss.) Roem*), paulownia wood (*Paulownia fortunei*), beech wood (*Zelkova serrata (Thunb.) Makino*), ashtree (*Fraxinus mandschurica Rupr*.), black walnut wood (*Juglans nigra* L.), rosewood (*Ormosia henryi Prain*) with 50 mm in length and width, 2 mm in thickness. The serial numbers are OW-A, OW-B, OW-C, OW-D, OW-E, OW-F, OW-G, and OW-H, all purchased from Yihua Life Technology Co., Ltd., Shantou, China. Ethyl orthosilicate (TEOS) was purchased from Chengdu Kelong Chemical Reagent Factory, Chengdu, China. The ammonia (NH_3_·H_2_O, 28%), sodium hydroxide (NaOH,96%), and hydrochloric acid (HCl, 36~38%) were provided by Nanjing Chemical Reagent Co., Ltd., Nanjing China. The anhydrous ethanol (99.7%) was from Sinopsin Chemical Reagent Co., Ltd. Shanghai, China, and 1H,1H,2H,2H-perfluorodecyl trichlorosilane (FDTS) purchased from Aladdin Shanghai Co., Ltd., Shanghai, China. Coffee, milk, and tea were purchased from the local supermarket. Distilled water was provided by our laboratory.

### 2.2. Preparation of Silica Micro–Nano Coating on Wood Surface

The poplar wood used in this experiment was ultrasonically cleaned (Misonix, Inc., New York, NY, USA) in anhydrous ethanol for 30 min and distilled water for 30 min, dried to absolute dry in a constant temperature blast oven (Shaoxing Shangchen Instrument Co., Ltd., Shaoxing, China) at 60 °C, then removed and sealed for use. Add 200 mL deionized water to 240 mL ethanol. Then 40 mL ethyl orthosilicate (TEOS) was quickly dropped into the ethanol and water mixture and magnetically stirred for 15 s. Under stirring conditions, 4 mL ammonia water was dropped into the mixture, continued stirring for 3 h, and then stood for 24 h to obtain light blue silica sol. (Same steps, a total of 4 bottles). OW-A, OW-B (1 group); OW-C and OW-D (2 groups); OW-E and OW-F (3 groups). Samples of OW-G and OW-H (4 groups) were dipped into silica sol in 4 beakers and magnetically stirred at room temperature for 3 h to make SiO_2_ nanoparticles deposited on the surface of wood. Then, dry the wood in an oven at 85 °C for 2 h. The silica nanoparticles synthesized on the wood surface by the above method are rich in hydroxyl, and the wood surface is hydrophilic.

### 2.3. Modification of Silica Micro–Nano Coating on Wood Surface by FDTS

In this study, the surface of wood was modified with a silica micro–nano coating using vapor deposition. First, eight dried wood samples were placed in sealed containers containing approximately 0.2 mL of FDTS reagent and dried in a 120 °C oven for 2 h, enabling complete reaction between the silyl groups in the FDTS reagent and the hydroxyl groups of the silica nanoparticles. The samples were then transferred to a separate clean open container and dried in an oven at 130 °C for 3 h to remove any unreacted FDTS through volatilization, resulting in a hydrophobic wood surface. This method was used in all eight samples to ensure consistent modification. The modified poplar wood, elm wood, Chinese toon wood, paulownia wood, beech wood, ashtree, black walnut wood, and rosewood samples were, respectively, marked with MW-A, MW-B, MW-C, MW-D, MW-E, MW-F, MW-G, and MW-H. The preparation process is shown in Figure 1. The serial numbers of wood before and after modification are shown in Table 1.

### 2.4. Wood Density, Porosity, and Surface Roughness Test

The volume of unpainted wood was measured by the drainage method described in the national standard GB/T1933-2009 “Method for Determination of Wood Density”. The resulting measurement was precise to an accuracy of 0.001 cm^3^. The quality of the sample was determined to an accuracy of 0.001 g. The basic density of the sample was subsequently calculated using Formula (1), with a precision of 0.001 g/cm^3^.
(1)$$\rho_{0}=\frac{m_{0}}{V_{max}}$$

$\rho_{0}$ is the basic density of the sample, expressed in grams per cubic centimeter (g/cm^3^); Vmax is the volume of the sample when it is saturated with water, expressed in grams (cm^3^).

The quantitative analysis of macropore porosity and pore size parameters of wood was conducted using the AutoPore IV 9500(Micromeritics, Inc., Norcross, Georgia, NG, USA) mercuric injection instrument.

The JB-4C surface roughness tester, manufactured by Shanghai Gaozhi Precision Instrument Co., Ltd., Shanghai, China, was utilized to quantify the surface roughness of the wood samples. The wood surface roughness was measured in compliance with the wood surface roughness measurement method outlined in GB12472-90. The contour curve was captured perpendicularly to the direction of the wood grain, with a measurement length of 0.8 mm. The surface of each wood sample was measured four times at different locations, and the average value of the four measurements was taken.

### 2.5. Surface Morphology and Wettability Analysis

Use a microtome to cut the sample into 5 mm × 5 mm × 3 mm slices and stick them on the observation table through a conductive adhesive. The surface morphology was observed under vacuum by Quanta 200 environmental scanning electron microscopy (FEI Corporation, New York, NY, USA) (Hong, Sunghwan). The modified wood’s water contact angle (WCA) was measured by the Theta T200 optical contact angle analyzer (Biolin Technologies GmbH, Gothenburg, Sweden). The droplet volume was 6μL. The test was repeated three times on the surface of each sample to achieve the average value.

### 2.6. Surface Functional Group Test

Vertex 80 V Fourier transform infrared spectrometer (Brock Spectral Instruments GmbH, Karlsruhe, Germany) was used to analyze the chemical composition on the surface of the modified wood sample, with the wavenumber of 500~4000 cm^−1^.

### 2.7. Pollution Resistance and Self-Cleaning Ability Test

In order to evaluate the resistance of modified wood surfaces against pollution, a series of tests were conducted using contaminated liquids such as water, tea, coffee, milk, and methylene blue. Graphite was utilized as a surrogate for pollutants commonly encountered in daily life. The samples were positioned at a 30° angle from the horizontal plane on the edge of a Petri dish. A uniform layer of graphite powder was applied onto the sample surface, following which water droplets were vertically dispensed onto the surface using an eyedropper. The aim was to observe the self-cleaning ability of the sample under test conditions.

### 2.8. Mechanical Wear Resistance Test

The present study aimed to evaluate the wear resistance of the modified sample. To this end, a weight of 200 g was applied on the surface of the specimen, which was subsequently dragged 10 cm horizontally on a 400-mesh sandpaper. Subsequently, the sample was returned to its original position, completing a friction cycle. The experiment was conducted for 5, 10, and 20 cycles, respectively, while ensuring that the sandpaper remained in close contact with the surface of the coating during the entire duration of the friction cycle. The water contact angle of the sample surface was measured, and the average value of three measurements was calculated.

### 2.9. Water Resistance and Acid-Base Resistance Test

In order to determine the surface water resistance of the modified wood samples, slices of the prepared 8 kinds of wood samples were dipped in deionized water with the hydrophobic surface facing down, as shown in Figure 2. After 5 days at room temperature, the samples were taken out and dried at 85 °C for 1 h. The mass changes were measured and recorded with an electronic balance. In order to compare the water absorption of wood before and after modification, the water absorption test of wood before modification was added. The water absorption of the samples before and after modification was calculated by Formula (2).
(2)$$WA(\%)=(M_{\mathrm{ai}}-M_{\mathrm{bi}})/M_{\mathrm{bi}}\times 100\%$$

Among them, Mai is the weight after water absorption and Mbi is the weight of dry wood.

The acid and alkaline resistance of the modified wood were tested. The samples were soaked in hydrochloric acid solution with PH = 1 and sodium hydroxide solution with PH = 12. The samples were taken out every 24 h with the hydrophobic surface facing down, then it was washed with distilled water and dried at 120 °C for 2 h. Finally, the surface water contact angle (WCA) was measured and the average value of the three measurements was calculated.

## 3. Results and Discussion

### 3.1. Analysis of Wood Density, Porosity, Roughness, and Wettability

During the process of coating furniture surfaces in actual production, various quality defects may arise, including loss of luster, cracking, and detachment. These issues are not solely attributable to the coating process but are also closely linked to the inherent characteristics of the wood, such as moisture content, porosity, surface roughness, hardness, extractive content, and glossiness [22]. However, the specific ways in which these characteristics affect the finishing effect have not been reported in detail. Further research is needed to establish a more comprehensive understanding of the relationship between wood characteristics and the achievement of an ideal finishing effect in coating furniture surfaces. With the continuous improvement of people’s requirements on the function and decoration of furniture, it is of great significance to study the finishing performance of furniture, a vital wood property index, for improving the finishing quality of furniture, expanding the range of use and meeting people’s requirements for furniture [23]. In conjunction with the aforementioned experiments and findings, an analysis was conducted on eight types of natural wood in order to investigate the correlation between natural wood and superhydrophobic coating. The results presented in Table 1 demonstrate that the disparities in density, porosity, and roughness of the test wood specimens manifest the impact of wood’s intrinsic characteristics on the contact angle subsequent to surface coating.

Generally, wood density is small, pores are large, and coated coating adhesion is good; however, wood has higher density, fewer voids, and poor adhesion of coating [24]. As shown in Table 1, poplar wood (OW-A), elm wood (OW-B), Chinese toon wood (OW-C), and paulownia wood (OW-D) have soft materials, low density (0.312~0.680 g/cm^−3^), and relatively many pores, which provide a good environment for the deposition and arrangement of SiO_2_ micro–nano rough structures. After FDTS modification, it exhibited excellent superhydrophobicity. Beech wood (OW-E), ashtree (OW-F), black walnut wood (OW-G), and rosewood (OW-H) have hard materials, high density (0.693~0.794 g/cm^−3^), fewer pores, and poor environment of SiO_2_ micro–nano rough structure. The water contact angle of beech wood (OW-E) and rosewood (OW-H) is lower than that of other woods, and the superhydrophobic effect is not realized.

The roughness of a wood surface is determined by the spacing and height differences between its grooves and ridges, grooves, and holes, as well as grooves and protrusions [25]. The black lines in Figure 3 reflect the relationship between the original wood roughness and hydrophobicity. In cases where the substrate’s surface roughness is minimal, the mechanical interlocking between the coating and substrate is weakened, thereby leading to inadequate adhesion. Conversely, when the substrate’s surface roughness is substantial, it may damage the continuity and evenness of the coating, thereby hampering the coating’s adherence to the substrate surface [26]. The presented Table 2 indicates that beech wood (OW-E) and rosewood (OW-H) exhibit a significantly lower roughness, measuring 2.856 μm and 2.141 μm, respectively, than the remaining six types of woods. Conversely, paulownia wood (OW-D) displays the highest roughness, reaching 12.244 μm. Applying the Wenzel model (COSθ* = r COSθ_e_), where θ* denotes the contact angle of a rough surface under Wenzel condition, θ_e_ represents the intrinsic contact angle of the surface material, and r stands for the ratio of the actual solid/liquid interface contact area to the apparent solid/liquid interface contact area, reveals that when θ_e_ < 90°, θ* decreases with increasing surface roughness, whereas when θ_e_ > 90°, θ* increases with growing roughness. Following the introduction of SiO_2_ nanoparticles, the wood’s surface roughness increases, and θ_e_ > 90°. Therefore, with the increase in surface roughness, the micro–nano rough structure of the wood surface increases, the peaks and troughs in the roughness curve become larger, the single peak spacing becomes larger, and the average diameter of grooves and protrudes on the material surface increases. The actual solid/liquid interface contact area and the apparent solid/liquid interface contact area also increased, and the wood showed excellent superhydrophobicity. Beech wood and rosewood display a smoother roughness curve, smaller single peak spacing, smaller grooves and protrudes, and less micro–nano rough surface structure than the other types of woods. Consequently, their contact angles are smaller, failing to reach 150°. It is worth noting, however, that the contact angle does not increase infinitely with an increase in roughness. Rather, when the surface roughness of wood exceeds a critical value, the contact angle decreases, as evident in paulownia wood and ashtree. The red line in Figure 3 reflects the roughness ratio between the modified wood and the original wood. When the roughness ratio is at the minimum of 1.295, the wood achieves superhydrophobic.

### 3.2. Surface Morphology and Wettability Analysis

The SEM images presented in Figure 4 illustrate the low, high, and partial magnifications of the eight types of wood after FDTS modification. As observed in Figure 4(a1)–(h1), wood is a heterogeneous and porous material containing numerous grooves and voids on its surface. The application of silica nanoparticles on the wood surface results in their deposition and filling of the grooves. The nano-silica particles form a chemical bond between the hydroxyl group and the surface group on the wood surface, enabling their adhesion to the surface [27]. The results indicate that a layer of rough micro–nano protrusions, made of silica nanoparticles, was generated on the surface of eight types of wood after treatment, as evidenced by a2~h2 and locally enlarged images. These protrusions were distributed irregularly, without significant aggregation, but produced many voids and increased the surface roughness of the wood. This rough surface structure, combined with low surface energy FDTS, led to the entrapment of a significant amount of air in the grooves of the wood surface. As a result, when water contacts the wood surface, it initially encounters the air cushion, which prevents water droplets from wetting the wood, resulting in the wood exhibiting superhydrophobicity [28].

Table 3 and Figure 5 show the water contact angle values, standard deviations, and contact angle images of eight kinds of wood before and after modification. The size of the contact angle formed by water droplets on the wood surface provides a measure of its hydrophobicity. A contact angle of less than 90° is usually considered hydrophilic, while a contact angle greater than 90° is indicative of hydrophobic behavior. If the contact angle of water droplets on the wood surface exceeds 150°, the surface is classified as superhydrophobic [29]. When water droplets were placed on the surface of eight untreated wood samples, the droplets immediately spread out on the surface, exhibiting a water contact angle close to 0°, which is indicative of good wettability and confirms that wood is a hydrophilic material. However, after depositing silica nanoparticles and modifying the wood surfaces with FDTS, the contact angles for poplar (MW-A), elm (MW-B), Chinese toon (MW-C), paulownia (MW-D), ashtree (MW-F), and black walnut (MW-G) woods were measured to be 161.9°, 154.2°, 156.4°, 152.3°, 152.0°, and 165.2°, respectively. This transformation from hydrophilic to hydrophobic was observed through experiments where water droplets rolled off the surface easily, leaving it completely dry. While beech wood (MW-E) at 148.2° and rosewood (MW-H) at 146.4° did not reach the 150° threshold required for superhydrophobic properties, they still demonstrated good hydrophobic properties.

The excellent superhydrophobic characteristics of wood surface can be attributed to the combined effects of increased roughness of spherical SiO_2_ nanoparticles and the hydrophobic modifier perfluorodecyl trichlorosilane (FDTS). As depicted in Figure 4, SiO_2_ nanoparticles are uniformly distributed and tightly arranged on the wood surface, resulting in a rough structure after FDTS modification. The long chains of hydrophobic alkane of FDTS are successfully attached to the surface of SiO_2_ nanoparticles, thus facilitating the attainment of superhydrophobic properties of wood. Figure 6 depicts the process of creating superhydrophobic properties on the wood surface through the deposition of silica nanoparticles on the wood surface, followed by modification with FDTS reagent. The wood surface treated with silica sol contains a significant number of hydroxyl groups on the surface of SiO_2_ nanoparticles, resulting in strong hydrophobicity. Upon surface modification with FDTS, the presence of long-chain alkyl molecules provides increased hydrophobicity, which imparts excellent superhydrophobic ability to the wood surface.

### 3.3. Analysis of Reaction Mechanism

The infrared spectrum of the modified wood is presented in Figure 7, which displays several characteristic absorption peaks in the wave number range of 500 to 4000 cm^−1^. The absorption band at 3332 cm^−1^ is attributed to the stretching vibrations of the -OH group. Meanwhile, the bands at 2920 cm^−1^ and 2853 cm^−1^ are due to the asymmetric stretching vibration of -CH_3_ and symmetric stretching vibration of -CH_2_- in the long chain of alkyl molecules of FDTS, respectively. The absorption peak at 1735 cm^−1^ corresponds to the stretching vibration of C=O. In addition, the bands at 1031 cm^−1^ (1025 cm^−1^), 897 cm^−1^, and 446 cm^−1^ are the result of the asymmetric stretching vibration, symmetric stretching vibration, and bending vibration of Si-O-Si, respectively [30,31]. The absorption bands observed at 1350~1000 cm^−1^ and 800 ~ 600 cm^−1^ correspond to the stretching vibrations of C-F and C-Cl, respectively. The successful grafting of FDTS onto the surface of SiO_2_ particles was confirmed by the hydrolysis and chemical reaction of FDTS with the hydroxyl groups present on the SiO_2_ surface. Notably, MW-E and MW-H exhibited intense bands at 1203 cm^−1^ and 704 cm^−1^, corresponding to the CF_2_ and CF_3_ stretching vibrations and the CH_2_-Cl stretching vibration, respectively, both of which are attributed to FDTS. These two kinds of wood exhibited different FDTS modification absorption peaks compared to other woods, indicating that the micro–nano rough structure of MW-E and MW-H had distinctive effects on FDTS modification. Although this difference could not be directly observed through SEM images, the water contact angle results confirmed the difference in the modification. The contact angle for beech wood (MW-E) and rosewood (MW-H) was 148.2° and 146.4°, respectively, both of which did not reach 150°. The roughness and hydrophobicity of wood surface coating are mainly provided by modified silicon dioxide. Natural beech wood (2.856 um) and rosewood (2.141 um) have low roughness and contain less modified silicon dioxide, which cannot provide suitable roughness. The modification of low surface energy FDTS is limited, and beech wood and rosewood fail to achieve a super hydrophobic effect [32].

### 3.4. Analysis of Contaminant Resistance Test and Self-Cleaning Test

Figure 8 and Figure 9 depict the experimental evaluation of pollution resistance and self−cleaning of modified wood, respectively. As depicted in Figure 8, a variety of pollutants were applied to the surface of the modified wood for a duration of five minutes. The results indicate that upon wiping, no residual traces were observed, which is indicative of low adhesion and high mobility [33]. This suggests that the coating exhibits a robust resistance to these pollutants, thereby increasing its durability and prolonging its useful life. In Figure 9, a self−cleaning test was performed on the modified wood by applying graphite powder to its surface to simulate real-life contaminants, such as dust. The droplets were then released onto the coating from a 45° angle. Due to the low adhesion of the wood surface, the droplets were observed to roll off the surface, taking along the pollutants with them. Subsequently, the droplets containing the pollutants were seen to depart from the surface of the coating under gravity, thereby achieving a self−cleaning effect of the material. These observations provide compelling evidence that the modified wood possesses desirable properties of self−cleaning and antifouling, making it suitable for application in industries such as furniture and construction.

### 3.5. Mechanical Wear Resistance Analysis

Figure 10 shows the effect of the number of sanding cycles on the change in the water contact angle of the modified wood. The average water contact Angle and standard deviation of 5, 10 and 20 mechanical wear tests are shown in Appendix A. With the increased number of sanding cycles, the water contact angle of the modified wood surface decreased. Specifically, the WCA of unsanded samples ranged from 146.4° to 165.2° for different types of wood. After five sanding cycles, the WCA decreased by 3% to 18% compared to the unsanded samples. The maximum reduction in WCA was observed for black walnut wood (21%), while rosewood showed the smallest decrease (8%) after 10 sanding cycles. After 20 sanding cycles, the WCA of black walnut wood decreased by 27%, and that of ashtree decreased by 13%. Figure 10b displays the SEM image of ashtree and black walnut wood after undergoing 20 cycles of friction. The image reveals that the sanding has led to the erosion of the abrasive surface structure, resulting in a smoother surface. Additionally, the low surface energy material, FDTS, has been consumed by the abrasive silica structure. As a result of the modification, the contact area between the modified wood surface and water droplets has increased, leading to a decline in its hydrophobic nature. The superhydrophobic properties of the modified wood surface were lost after 20 sanding cycles, but the hydrophobic properties still persisted, indicating weak wear resistance of the superhydrophobic coating. Therefore, future research should focus on enhancing the wear resistance of the superhydrophobic coating to prolong its superhydrophobic properties.

### 3.6. Water Resistance Analysis

The absorption of water by wood over extended periods, whether due to wet conditions or washing, can lead to substantial water retention, which causes the wood to undergo dimensional deformation or decay. The water absorption of wood severely limits the application of wood products. Therefore, improving wood's hydrophobic ability can effectively prevent wood dimensional deformation and delay decay. Figure 11 shows the results of eight natural and modified wood samples after 120 h of water immersion. The modified wood samples exhibit substantially lower water absorption rates than the untreated wood. Among the modified samples, elm wood (B) shows the highest reduction in water absorption rate, with a decrease from 40.3% to 5.6% after treatment. Poplar wood (A), on the other hand, displays the smallest reduction in water absorption, with values decreasing from 27.8% to 19.2% after treatment. However, the modified wood samples still exhibit a certain degree of absorbency, likely attributable to the abundant microporous structures, such as cell cavities and grooves, present in the wood. A small amount of water is initially absorbed through capillary action, followed by saturation of the entire sample. During the experiment, only one side of the wood was modified to be hydrophobic, while the rest of the wood remained uncoated. The porous structure of the wood is responsible for water absorption. The modified wood surface, with its rough texture and low surface energy, displays certain water-repellent properties, thereby reducing the wood’s water absorption capacity. These findings suggest that superhydrophobic modification of wood renders it suitable for use in humid environments.

### 3.7. Acid and Alkali Resistance Analysis

The effect of impregnation time with a hydrochloric acid solution (pH = 1) on the surface wettability of modified wood is presented in Figure 12a. The mean water contact Angle and standard deviation of acid resistance test are shown in Appendix A. As depicted in the figure, the contact angle between the modified wood surface and water decreases with an increase in impregnation time in the hydrochloric acid solution. The largest decrease (24%) in contact angle was observed for paulownia wood after 72 h of immersion, while the smallest change (14%) was noted for elm wood. The study findings indicated that silica nanoparticles and 1H,1H,2H,2H-perfluorodecyl trichlorosilane (FDTS) are vulnerable to long-term exposure to strong acids. The anti-corrosion properties of the superhydrophobic surface can be attributed to the formation of a composite interface between the micro–nano rough structure of the surface and air and water, thereby reducing contact between the interface and the wood surface [34]. However, prolonged immersion in an acidic solution can lead to the corrosion of the composite interface by corrosive ions such as Cl^−^, thereby damaging the interface to a certain extent [35]. Consequently, silica particles fall off and substances with low surface energy are decomposed, which negatively affects the hydrophobic effect of the modified wood.

The present figure, namely Figure 12b, illustrates the impact of immersion time of sodium hydroxide solution, at a pH of 12, on the surface wettability of modified wood. The mean water contact Angle and standard deviation of acid resistance test are shown in Appendix A. As observed in the figure, the contact angle between the surface of the modified wood and water decreases with the increase in immersion time in sodium hydroxide solution. Following 72 h of dipping, the contact angle of the eight samples was reduced by 19% for black walnut wood and 10% for rosewood. It is notable that the coating is influenced by a certain concentration of alkali solution, which results in the reduction of the hydrophobic property. Nevertheless, even after 72 h of immersion in sodium hydroxide solution, the water contact angle of the modified wood surface remains above 120°, and the coating maintains its hydrophobic nature, indicating a certain degree of resistance to alkali solution corrosion. During the impregnation of the alkaline solution, the fluorosilanes in 1H,1H,2H,2H-perfluorodecyl trichlorosilane (FDTS) undergo a breakage to form hydrophilic hydroxyl groups. When wood is subsequently heated, the hydrolyzed molecules recombine, and the surface becomes hydrophobic once again [36].

Compared with the two figures, the contact angle of an alkaline solution tends to be greater than that of an acidic solution subsequent to immersion. Based on the Cassie and Baxter model, as well as extensive research, the contact between a water droplet on a superhydrophobic surface and a solid surface comprises two forms of contact, namely, solid–liquid contact and liquid–gas contact. In an acidic environment, there is some direct interaction between an acidic solution and modified wood, which results in the breaking of hydrogen bonds between molecules [37]. Additionally, acid or alkali solutions can corrode and dissolve some of the silicon dioxide in the coating, as well as the low surface energy silane on the surface, which may spall off, causing the original rough structure to deteriorate and a reduction in surface hydrophobicity. The findings of this study indicate that the FDTS-modified surface possesses some degree of resistance to alkaline corrosion but is prone to damage in an acidic environment.

## 4. Conclusions

The current study successfully fabricated SiO_2_-FDTS superhydrophobic coatings on various wood substrates via sol–gel and vapor deposition techniques. Silica sol was impregnated into the wood to create a micro–nano rough structure, followed by grafting with low surface energy perfluorodecyl trichlorosilane to yield six types of superhydrophobic coatings and two types of hydrophobic coatings. To examine the correlation between the natural wood properties and the superhydrophobic coating, we evaluated the density, porosity, and roughness of eight types of natural wood. The results indicated that the wood with low density, high porosity, and roughness provided an optimal environment for the silica deposition and arrangement. Furthermore, the coated wood demonstrated exceptional self-cleaning, anti-fouling, and water-repellent properties. After 72 h of immersion in acid and alkali solutions, the coating exhibited excellent resistance to alkali solution corrosion and moderate resistance to acid corrosion. Following 20 rounds of friction with 400-mesh sandpaper, the superhydrophobic property was lost; nevertheless, the water contact angle remained at 120°, indicating the presence of the hydrophobic effect. This research provides valuable insights into the hydrophobic modification of various wood substrates, which could potentially promote the use of superhydrophobic coatings in the furniture industry. Future research should aim to enhance the coating properties and expand the application scope of superhydrophobic coatings.

## Figures and Tables

**Figure 1 polymers-15-01644-f001:**
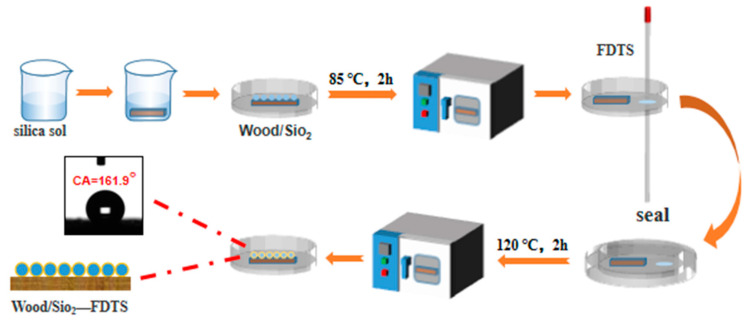
Preparation process of superhydrophobic wood.

**Figure 2 polymers-15-01644-f002:**
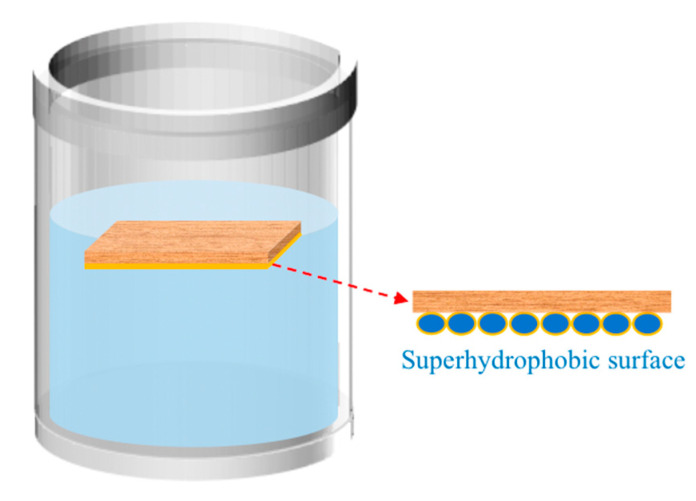
Demonstration of impregnation of superhydrophobic wood.

**Figure 3 polymers-15-01644-f003:**
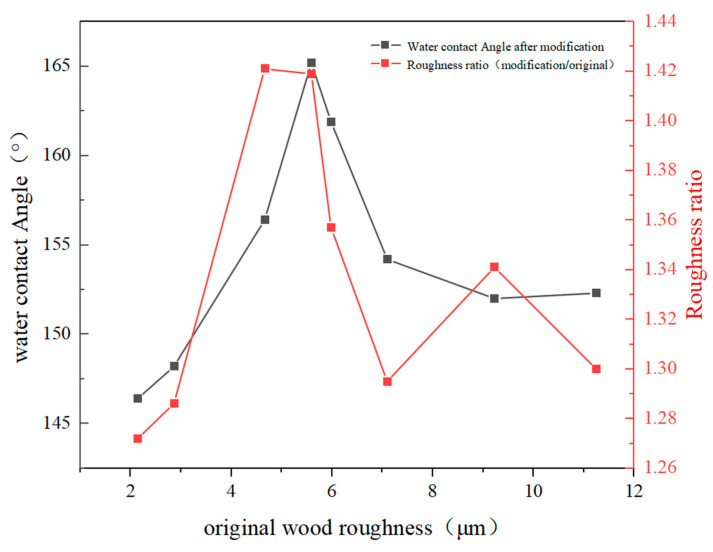
The relationship between original roughness and hydrophobicity and the roughness ratio of wood before and after modification.

**Figure 4 polymers-15-01644-f004:**
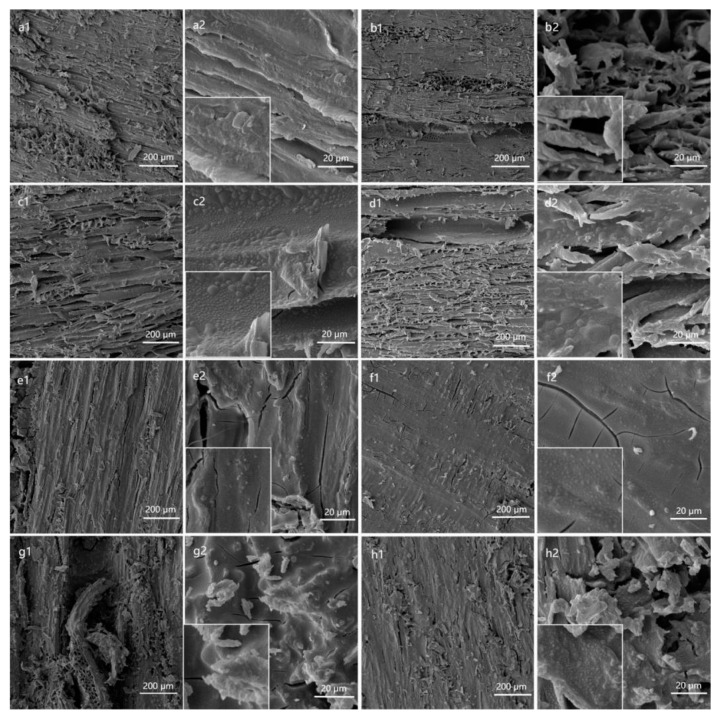
SEM image of superhydrophobic coating surface of eight kinds of wood. (**a** Poplar wood, **b** elm wood, **c** Chinese toon wood, **d** paulownia wood, **e** beech wood, **f** ashtree, **g** black walnut wood, **h** rosewood) low magnification, high magnification and locally enlarged image.

**Figure 5 polymers-15-01644-f005:**
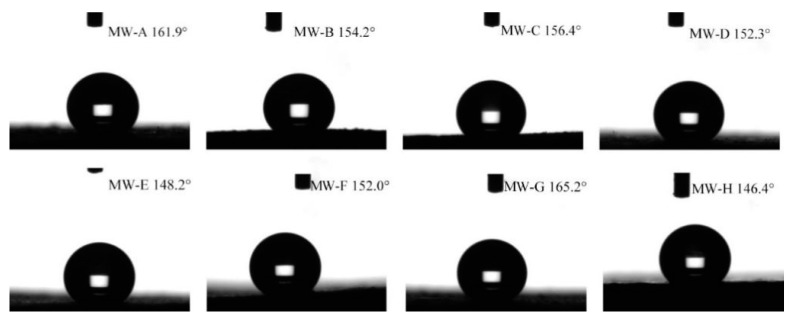
Water contact angle on the modified wood surface.

**Figure 6 polymers-15-01644-f006:**
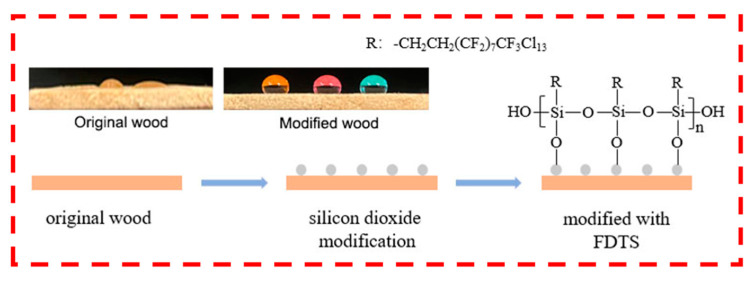
The forming processes of superhydrophobic wood surface. R represents the hydrophobic group.

**Figure 7 polymers-15-01644-f007:**
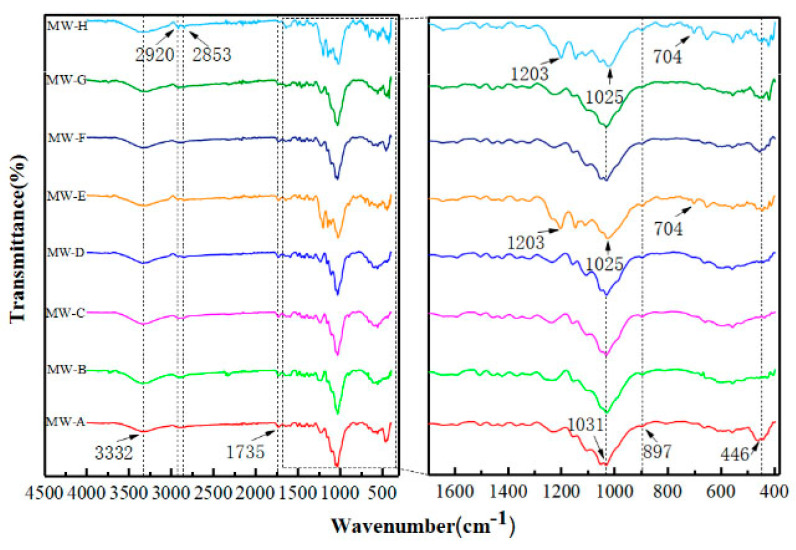
FTIR spectra of modified wood.

**Figure 8 polymers-15-01644-f008:**
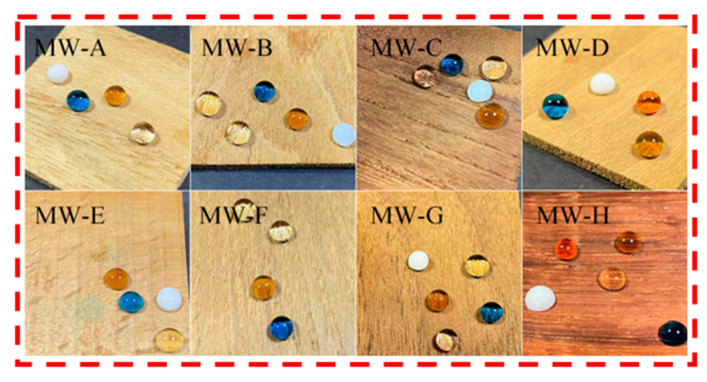
Different pollutants on modified wood (water, tea, coffee, milk, methylene blue).

**Figure 9 polymers-15-01644-f009:**
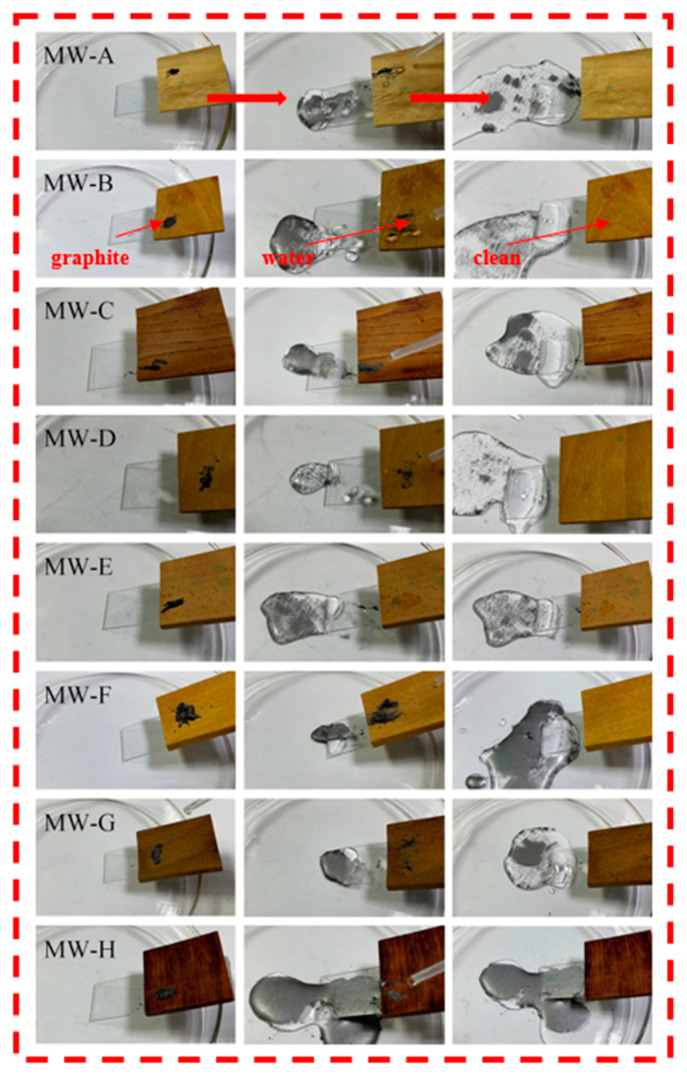
Self-cleaning test of modified wood.

**Figure 10 polymers-15-01644-f010:**
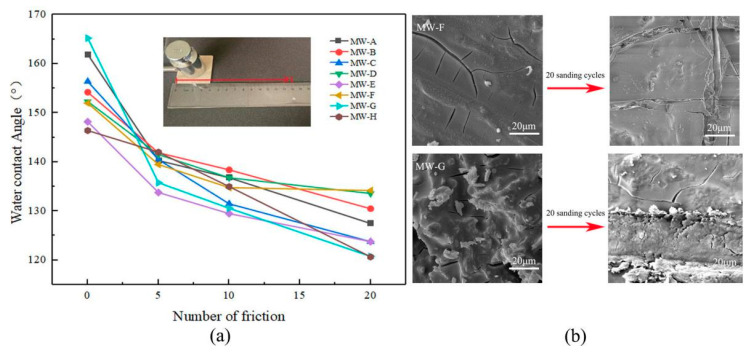
(**a**) Wear resistance test of modified wood. (**b**) SEM image of ashtree and black walnut wood after 20 times of friction.

**Figure 11 polymers-15-01644-f011:**
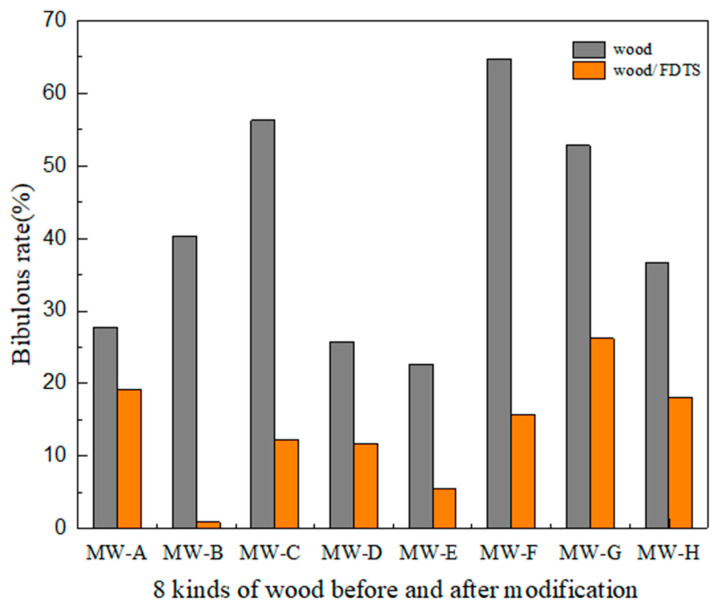
Water resistance test of untreated and modified wood.

**Figure 12 polymers-15-01644-f012:**
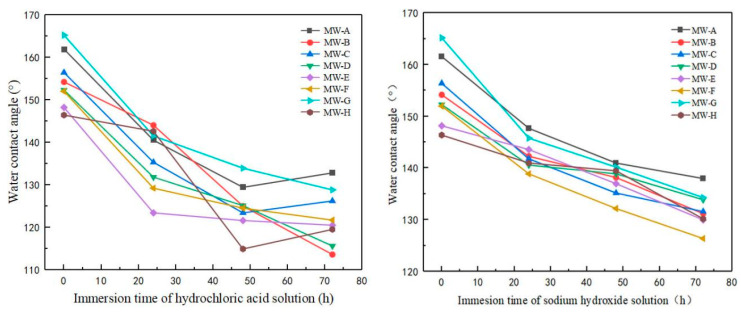
Acid and alkali resistance test of modified wood (**a**) HCl solution (pH = 1) and (**b**) NaOH solution (pH = 12).

**Table 1 polymers-15-01644-t001:** Serial numbers of wood before and after modification.

Wood	Poplar Wood	Elm Wood	Chinese Toon Wood	Paulownia Wood	Beech Wood	Ashtree	Black Walnut Wood	Rosewood
Original wood	OW-A	OW-B	OW-C	OW-D	OW-E	OW-F	OW-G	OW-H
Modified wood	MW-A	MW-B	MW-C	MW-D	MW-E	MW-F	MW-G	MW-H

**Table 2 polymers-15-01644-t002:** Original wood density, porosity, and roughness.

Sample	Density (g/cm^−3^)	Porosity (%)	Roughness (um)
OW-A	0.386	29.81	5.978
OW-B	0.680	13.66	7.093
OW-C	0.540	27.36	4.660
OW-D	0.312	42.36	11.244
OW-E	0.693	14.54	2.856
OW-F	0.794	30.27	9.222
OW-G	0.730	23.33	5.586
OW-H	0.722	22.23	2.141

**Table 3 polymers-15-01644-t003:** Wood average water contact angle and standard deviation before and after modification.

Original Wood			Modified Wood		
Serial Number	Average Water Contact ANGLE	Standard Deviation	Serial Number	Average Water Contact ANGLE	Standard Deviation
OW-A	3.2°	1.08	MW-A	161.9°	3.08
OW-B	12.6°	2.36	MW-B	154.2°	1.15
OW-C	15.4°	2.21	MW-C	156.4°	3.91
OW-D	7.5°	1.68	MW-D	152.3°	3.79
OW-E	10.6°	2.58	MW-E	148.2°	4.33
OW-F	2.1°	3.14	MW-F	152.0°	5.34
OW-G	4.5°	2.16	MW-G	165.2°	2.34
OW-H	5.2°	1.45	MW-H	146.4°	3.89

## Data Availability

The data presented in this study are available on request from the corresponding author.

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
