# Peer review of "Study on the Difference of Superhydrophobic Characteristics of Different Wood Furniture Substrates"

_polymers, 2023, doi:10.3390/polym15071644_

Round 1

Reviewer 1 Report

This manuscript describes the method of creating a superhydrophobic protective layer on a wood surface using fluorinated silica. The coating performance on different wood species has been studied through characterizing the surface properties and varying the external condition. More experiments and data analysis are needed to support the conclusions.

1. What are the water contact angle values of different wood species before treatment? These control groups need to be provided.

2. For each sample, water contact angle needs to be measured at multiple spots and the statistical results are required (average and standard deviation) given that wood surface texture varies from spot to spot. Likewise, scale bars are needed for plotting Figure 9 and 11.

3. A quantitative relation between surface roughness and hydrophobicity should be provided by correlating data in Table 1 and Figure 4 to better support the findings.

4. If author claim that “abrasive wear of the surface structure caused by sanding”, microscope imagines should be provided to show the surface structural change.

5. In section 3.6. water resistance analysis, how to correlate the microporous structures of different wood species (like SEM pictures) to their water absorption rate before/after modification?

6. A minor comment: Section 2.6, and 2.7 have the repeated title as 2.5, which are not consistent with the content in the corresponding paragraphs.

Reviewer 2 Report

The authors present their study on developing a superhydrophobic silica coating on various wood surfaces. The coating was produced using a commonly used fluorinated functional group modification with silica sol. They noted the lack of recent research on superhydrophobic coatings for wood surfaces, which led them to investigate the relationship between different surface characteristics, material properties, and coating development. The experiments thoroughly analyzed the material characteristics and performance as superhydrophobic surfaces. The authors provide a comprehensive background introduction and reference search, and the experiments are presented in detail. However, the manuscript's novelty is limited due to the poor mechanical durability of the coating and the utilization of a method that has already been reported in the past few decades.

To my review, I would like to have these comments:

1. I suggest that creating a table to link different serial numbers with the different wood types used in Experiment 2.1 would improve clarity for the reader.

2. It would be helpful to know if the treatments (water/EtOH dipping, heating) applied to the wood in preparation for the section 2.2 affected the material surface properties. Addressing this point could provide additional insight into the results and add value to the study.

3. Tile of section 2.6, 2.7 needed to be double check which right now not fit to the content.

4. As the author mentioned, the Wenzel model near the line 255.

First, in my understanding, the intrinsic angle should be FTDS on the flat surfaces (r=1), so what is this angle? Based on this intrinsic angle, what is the minimal roughness ratio needed to achieve superhydrophobic? Did author has the exact roughness ratio for each sample?

By the way, did the author think about the CB model that the air bubble will be shown in the superhydrophobic surfaces? Based in the CB model, what is the ratio between liquid-solid and solid-gas phase contact, considering the experimental contact angle and intrinsic angle.

5. Based on the SEM image, the coverage of SiO2 particles look different for various wood surfaces. If the silica not fully covered the surfaces, will the FDTS graft on the wood surfaces?

6. the manuscript's novelty lies in the author's attempt to develop a superhydrophobic silica coating on different wood surfaces and investigate the relationship between wood characteristics and coating development. This approach could provide new insights into the development of superhydrophobic coatings for wood surfaces and broaden the understanding of the influence of surface characteristics on coating performance.

But the novelty is not intense compared to their featureless coating resistance which is very similar to plenty of superhydrophobic coating that made by fluorinated compound modified silica surfaces.  These reports always display highly superhydrophobic properties as self-cleaning, anti-fouling, antiadhesion but with poor resistance especially mechanical durability.

Round 2

Reviewer 1 Report

The author's efforts to address each point have resulted in a much clearer and more coherent manuscript. The additional data and updated analysis have enhanced the validity and significance of the findings. Therefore, I am happy to recommend its publication.

Reviewer 2 Report

Thanks for addressing my questions and concern. I think the content now looks perfect right now. The only problem for me is the novelty since plenty of studies are about achieving superhydrophobicity with F-silane and silica nanoparticles.

As for the improved manuscript, I agree to publish the revised version.